# Identification of Quantitative Trait Loci Associated with Seed Protein Concentration in a Pea Recombinant Inbred Line Population

**DOI:** 10.3390/genes13091531

**Published:** 2022-08-26

**Authors:** Junsheng Zhou, Krishna Kishore Gali, Ambuj Bhushan Jha, Bunyamin Tar’an, Thomas D. Warkentin

**Affiliations:** Crop Development Centre, Department of Plant Sciences, University of Saskatchewan, 51 Campus Drive, Saskatoon, SK S7N 5A8, Canada

**Keywords:** pea, *Pisum sativum*, seed protein concentration, marker-assisted selection, QTLs

## Abstract

This research aimed to identify quantitative trait loci (QTLs) associated with seed protein concentration in a recombinant inbred line (RIL) population of pea and aimed to validate the identified QTLs using chromosome segment-introgressed lines developed by recurrent backcrossing. PR-25, an RIL population consisting of 108 F7 bulked lines derived from a cross between CDC Amarillo (yellow cotyledon) and CDC Limerick (green cotyledon), was used in this research. The RIL population was genotyped using an Axiom 90K SNP array. A total of 10,553 polymorphic markers were used for linkage map construction, after filtering for segregation distortion and missing values. The linkage map represents 901 unique loci on 11 linkage groups which covered a map distance of 855.3 Centimorgans. Protein concentration was assessed using near-infrared (NIR) spectroscopy of seeds harvested from field trials in seven station-years in Saskatchewan, Canada, during the 2019–2021 field seasons. Three QTLs located on chromosomes 2, 3 and 5 were identified to be associated with seed protein concentration. These QTLs explained 22%, 11% and 17% of the variation for protein concentration, respectively. The identified QTLs were validated by introgression lines, developed by marker-assisted selection of backcross lines for introgression of corresponding chromosome segments (~1/4 chromosome) harboring the QTL regions. Introgression line PR-28-7, not carrying any protein-related QTLs identified in this study, was 4.7% lower in protein concentration than CDC Amarillo, the lower protein parent of PR-25 which carried one identified protein-related QTL. The SNP markers located at the peak of the three identified QTLs will be converted into breeder-friendly KASP assays, which will be used for the selection of high-protein lines from segregating populations.

## 1. Introduction

The demand for plant-based proteins has been expanding rapidly over the past decade [1]. Plant-based proteins are used in many food applications, including in beverages and in meat analogues [2,3]. The current global plant-based protein market is worth USD 13.2 billion, and its projected compound annual growth rate is >10% [4]. Several driving forces have contributed to the increased attention on plant-based protein. The continuously growing global population creates a greater demand for proteins, while the production of animal proteins, subject to the scale and input, is unlikely to fulfill the needs of a growing population [5]. From an environmental perspective, progressive consumers want to choose food products with a smaller carbon footprint [6]; legume crops which naturally fix atmospheric nitrogen and improve soil fertility for subsequent crops are a perfect fit. From a nutritional perspective, plant-based proteins typically contain less fat and no cholesterol compared to meat protein, and so are attractive to consumers with an increased willingness to follow a healthy diet [7]. The emergence of improved meat analogues provides an option for meat lovers that is more favorable for their health without having to compromise on taste and texture.

Soybean and pea are the top two legumes available on the plant-based protein market [8]. Though pea accounted for a smaller share of the market compared to soybean, pea possesses the merits of low allergenicity, a less unpleasant flavor and a more affordable price, which are favored traits in the meat analogue industry [9]. As the demand has increased, pea production has increased prominently in the past five years. For example, the Food and Agriculture Organization (USA) reported a 57.5% increase in pea production in the United States in 2020 compared to 2017. The need for pea is continuously expanding as the estimated compound annual growth rate of the global pea market is 3.4% in the next 5 years [10]. The growth potential of the plant-based protein market, and the increasing demand for pea proteins, is encouraging many pea-breeding programs to focus on enhancing protein concentration to fulfill the food and ingredient demand [11,12]. Understanding the genetic basis of seed protein concentration is important to improve the quality of future pea varieties. QTL information has provided valuable tools for the improvement of several complex traits in crop plants through marker-assisted selection [13]. However, the information on the genetic basis of key traits is less well-known in pea compared to soybean and other crops [14,15]. The studies conducted on pea to understand the genetic basis of seed protein concentration have identified several associated QTLs that explained the phenotypic variance ranging from 4 to 22 percent [16,17,18,19]. In the current study, we used a mapping population derived from CDC Limerick, which is the cultivar with the highest seed protein concentration among the pea cultivars released by the Crop Development Center (CDC) over the last three decades.

Linkage mapping is a common method to reveal genomic regions associated with traits of interest and has been widely implemented in many breeding programs [20]. Recombinant inbred line (RIL) and backcross populations are often used as genetic resources for linkage mapping. RILs are ideal populations for QTL mapping and can reveal multiple loci that contribute to the trait of interest [21]. Backcross populations and introgression lines are a valuable resource for validating the effect of individual QTLs, since the background genetic variation is eliminated through recurrent backcrossing [22]. Introgression lines are also a tool for the fine mapping of identified QTLs [23]. Both inbred and backcross populations have been used for high-resolution mapping of protein-associated traits [24].

In the current study, 108 RILs of the PR-25 mapping population, developed at the Crop Development Centre, University of Saskatchewan, and derived from the cross between CDC Amarillo [25] and CDC Limerick [26], were used for linkage mapping and identification of QTLs associated with seed protein concentration. Chromosome segment-introgressed lines developed by recurrent backcrossing and marker-assisted selection were used to validate the effect of each identified QTL independent from each other, by introgressing the QTLs into different sister lines.

## 2. Materials and Methods

### 2.1. PR-25 Population

PR-25 is a recombinant inbred line (RIL) population with 108 progeny lines derived from a cross between CDC Amarillo and CDC Limerick. The maternal parent CDC Amarillo was developed by the Crop Development Centre (CDC), University of Saskatchewan from the pedigree CDC Golden/CDC 715-4//CDC Meadow/CDC 0108 [25]. CDC Limerick, used as the pollen donor, was developed from the pedigree CDC0107/PS 610152//CDC0007 [26].

The PR-25 population was grown at several locations in Saskatchewan, Canada from 2019 to 2021, including two replicates at Sutherland (near Saskatoon) in 2019, three replicates at Sutherland, Rosthern and Lucky Lake in 2020, and three replicates at Floral, Rosthern and Lucky Lake in 2021. Among these nurseries, Sutherland and Floral are located in the Dark Brown soil zone, Rosthern is located in the Black soil zone, and Lucky Lake is located in the Brown soil zone. Eighty-four seeds of each line were sown in 1 m^2^ micro-plots in a randomized complete block design at each station-year. PR-25 was managed using the best management practices for pea in western Canada throughout 2019–2021. Detailed information on the plant management, including seeding and harvest dates, is listed in Table 1.

### 2.2. Protein Concentration Assessment

Harvested seeds from each experimental plot were air dried to approximately 14% moisture content, then stored at the CDC pulse crop field lab at room temperature (22 °C) to equilibrate prior to near-infrared (NIR) spectroscopy analysis. NIR spectroscopy is a non-destructive method for the measurement of protein concentration as well as other proximate analyses in pea seeds. The R2 values for crude protein concentration in the CDC pea calibration [27] are greater than 0.95. The accuracy of the NIR calibration was monitored and enhanced by annual HPLC analyses of approximately 100 randomly selected pea seed samples grown in field trials in the previous season. A FOSS NIR Systems 6500 Near-Infrared Spectrophotometer (Foss Tecator, Hoeganaes, Sweden) was used at the Grains Innovation Lab, University of Saskatchewan, to collect spectra from the pea samples using the natural products transport cup. The reflectance of the pea samples from 400 to 2498 nm in 2-nm increments was obtained during each scan. Twenty-five scans were collected and averaged for each sample. The correlation equation developed by Arganosa et al. [27] was used for calculating the predicted crude protein concentration.

### 2.3. Agronomic Assessment

The phenotypic data of traits including the leaf type, the plant stand density (PSD), the flower color, the plant height, the lodging score, the Mycosphaerella blight (Myco), the number of days to flowering (DTF), the number of days to maturity (DTM), the seed yield, and the thousand seed weight (TSW) were collected. The DTF was determined as the number of days from the sowing to the flowering stage. The DTM was determined as the number of days from planting to physiological maturity. The TSW was determined as the weight in grams of 1000 seeds. The yield was measured at 14% seed moisture content.

### 2.4. Genotyping and Linkage Map Construction

The population was genotyped using an Axiom^TM^ 90K SNP array developed by INRA, France, based on the SNPs identified in a diverse panel of pea lines. The array was obtained from Thermofisher (Waltham, MA, USA) and array hybridization for genotyping was conducted by Euroffins (San Francisco, CA, USA). A total of 10,553 polymorphic SNP markers were used to construct a linkage map after filtering for missing values and segregation distortion. The markers were binned using IciMap and the bin representative markers were used for linkage map construction using MstMap. A logarithmic of odds (LOD) value of 10 and a missing value threshold of 10% were used for linkage mapping. A total of 3384 markers were mapped, and the linkage map represented 901 unique loci on 11 linkage groups which covered a map distance of 855.35 cM (Figure 1).

### 2.5. QTL Identification

The marker order presented in the linkage map and the phenotypic information collected during the seven trials were used for QTL mapping. Windows QTL Cartographer V2.5_011 software (Department of Statistics, North Carolina State University, Raleigh, USA) [28] was used to identify protein concentration-related QTLs. The composite interval mapping (CIM) method was used, along with the Kosambi’s mapping function. The LOD threshold value was set at 3.1 based on 1000 permutations and a significance level of 0.05. The walk speed was selected as 1.0 cM.

### 2.6. QTL Validation Using Introgression Lines

A 1/4-chromosome segment substitution line (CSSL) library of pea named PR-28 was developed at the Crop Development Centre, University of Saskatchewan [29]. The CSSL library contains a set of lines derived from selected BC4F2 individuals obtained by recurrent backcrossing of CDC Amarillo × CDC Limerick F2 lines. These lines were a subset of F2 lines used to develop the PR-25 RIL population. CDC Amarillo was used as the recurrent parent and KASP assays of 12–15 selected SNP markers positioned on each linkage group were used for foreground marker-assisted selection of chromosome segments in each backcross generation. Three selected CSSLs from this existing library, each carrying QTLs associated with seed protein concentration, were used for validation in the current study. In the genetic background of CDC Amarillo, PR-28-7 has introgressed a chromosome segment from 93.0 to 121.4 cM on LG1, PR-28-18 has introgressed a chromosome segment from 0 to 33.9 cM on LG3a, and PR-28-33 has introgressed a chromosome segment from 105.2 to 150.6 cM on LG5. The introgression lines were phenotyped for seed protein concentration in Sutherland in 2020, and in Floral, Rosthern and Lucky Lake in 2021. The plot design and phenotyping method were the same as what was used for the RIL population.

## 3. Results

The boxplots show the mean and the range of protein concentration in PR-25 in each station-year (Figure 2). Generally, the average protein concentration of PR-25 was higher in 2021 than in 2020. The variation in average protein concentration can be attributed to the differences in temperature and precipitation in each year. A higher summer temperature and lower precipitation were recorded in 2021 than in 2020, which resulted in a lower seed yield, but a greater protein concentration (Table 1). The boxplot of yield*protein showed that the total protein production in 2021, despite having the greater protein concentration, was significantly lower compared to 2019 and 2020 (Figure 3). ANOVA showed that both the genotype and station-year had significant effects on protein concentration (Table 2).

Three QTLs related to protein concentration were detected, and they were located on chromosomes 2, 3 and 5 (Table 3). The QTL (PC-QTL-1) was located on Chr2LG1 with the largest R2 value of 0.22; it ranged from 98.8 to 119.7 cM with a peak at 108.9 cM and its LOD score was 8.5 (Figure 3). The second QTL (PC-QTL-2) was located on Chr5LG3a with an R2 value of 0.11; it ranged from 13.1 to 22.2 cM with a peak at 20.2 cM and its LOD score was 4.83. The third QTL (PC-QTL-3) was located on Chr3LG5 with an R2 value of 0.17; it ranged from 124.6 to 144.7 cM with a peak at 139.7 cM and its LOD score was 5.71. The values of the additive effect for these three QTLs were 0.28, −0.20 and −0.25, respectively, indicating that PC-QTL-1 originated from CDC Limerick, while PC-QTL-2 and PC-QTL-3 originated from CDC Amarillo.

Protein concentration was negatively correlated with the seed yield and starch, similar to what has been reported in other pulse crops [30,31] (Table 4). The seed yield was positively correlated with the plant stand density and the plant height. The lodging score was negatively correlated with the seed yield, the plant height, the number of days to flower and the number of days to maturity, but was positively correlated with the Mycosphaerella disease score.

To avoid the possibility of unwittingly miscounting a seed yield-related QTL as protein-related, a QTL analysis was also conducted for the seed yield to distinguish the two. Only one QTL was found related to the yield and it was located on Chr5LG3a (Figure 1). Although PC-QTL-2 was also found on the same linkage group, they were on opposite ends of the chromosome; PC-QTL-2 was located from 18.1 to 20.7 cM, while yield-QTL-1 was located from 57.8 to 59.9 cM. Since there was no overlap between protein- and yield-related QTLs, the identified QTLs for protein concentration were not related to the yield trait.

Based on the means of 4 station-years, introgression line PR-28-7, which possesses none of the identified QTLs, had a lower protein concentration, while PR-28-18, which contains only PC-QTL-2, and PR-28-33, which contains only PC-QTL-3, had a higher protein concentration than CDC Amarillo and the average of all the PR-28 introgression lines (Table 5, Figure 4).

## 4. Discussion

Recombinant inbred line populations from bi-parental crosses are commonly used for linkage mapping and identification of QTLs [32]. PR-25 is an RIL population derived from the cross of CDC Amarillo and CDC Limerick. CDC Limerick has the highest protein concentration (26%) among all varieties developed by the Crop Development Centre; however, its green cotyledon color is less favored by the protein fractionation industries, which prefer yellow cotyledon pea varieties because they give rise to bright white protein fractions.

The primary focus of this study was to discover protein concentration-related QTLs, to facilitate the development of high-protein, high-yielding yellow pea varieties which are in demand from the growing plant-based protein market. CDC Amarillo is a high-yielding yellow pea variety with moderate protein concentration which is genetically distinct from CDC Limerick. The cross of CDC Amarillo and CDC Limerick generated sufficient genetic variation for the identification of QTLs associated with protein concentration. Some progeny lines with high yield, high protein concentration and yellow cotyledon color could be attractive to the protein fractionation industries. Several yellow pea progeny lines, including PR-25-61, PR-25-79 and PR-25-69, had greater protein concentration than the high-protein parent CDC Limerick across all station-years.

The use of an Axiom 90K SNP array in genotyping and linkage map construction resulted in high marker density and good coverage of unique loci across the pea genome, except for some centromere regions. The number of unique loci mapped on the PR-25 linkage map was greater than that of many of the recently published pea linkage maps [18,19]. The map distance of the PR-25 linkage map is on par with other pea linkage maps [18].

Seed protein concentration of the PR-25 population measured in seven station-years was used to identify the underlying QTLs. Using the mean values of seed protein concentration from these trials, three major QTLs located on chromosomes 2, 3 and 5 were identified in this study. The QTLs identified in this study were unique compared to the known QTLs for seed protein concentration in pea. We have compared the QTLs identified from previous studies with these current results by using the sequencing information of flanking markers of reported QTLs and used their base pair position on the pea genome sequence [33] for comparison. Gali et al. [18] reported a protein-related QTL on chromosome 2, with flanking markers Sc7251_83132 (Chr2LG1_7845408) and Sc2398_82962 (Chr2LG1_8771945), and a QTL on chromosome 5, with flanking markers Sc3132_175237 (Chr5LG3_217884075) and Sc3132_175238 (Chr5LG3_217884076). Klein et al. (2020) found a QTL on chromosome 3, with flanking markers PsCam034798_20158_236 (Chr3LG5_144456708) and PsCam012541_8505_720 (Chr3LG5_208141390) (Figure 5). This sequence-based comparison of flanking markers identified the protein-associated QTLs from this study (PC-QTL-1, PC-QTL-2, PC-QTL-3) as unique, because PC-QTL-1 had flanking markers Chr2LG1_291265214 and Chr2LG1_454521757, PC-QTL-2 had flanking markers Chr5LG3_15801800 and Chr5LG3_23895520, and PC-QTL-3 had flanking markers Chr3LG5_24108451 and Chr3LG5_437233435. PR-25 is one of the few published pea populations derived from a high-seed protein parent. The origin of the high protein concentration in CDC Limerick is likely different from parents of PR-02 and PR-07 [18], which are of moderate seed protein concentration, and the parents of Pop 10, as described by Klein et al. [19]. Therefore, PC-QTL-1, PC-QTL-2, and PC-QTL-3 appear to be novel QTLs associated with protein concentration.

PC-QTL-1 accounts for 22% of the phenotypic variation, which ranges from 98.8 to 119.7 cM on chromosome 2; PC-QTL-2 accounts for 11% of variation, which ranges from 13.1 to 22.2 cM on chromosome 5; and PC-QTL-3 accounts for 17% of variation, which ranges from 124.6 to 144.7 cM on chromosome 3. The loci within the peak region of each of these three QTLs, Chr2LG1_333534297, Chr5LG3_23895520 and Chr3LG5_424086163, could be used to develop markers associated with protein concentration and assist in the selection for future breeding projects.

The introgression lines PR-28-7, PR-28-18, and PR-28-33, selected from a chromosome segment substitution library, validated the protein concentration-associated QTLs identified in this study. PR-28-7, where its PC-QTL-1 region was masked by the introgression from CDC Limerick, was 4.7% lower in protein concentration compared to CDC Amarillo. In contrast, PR-28-18, which introgressed the PC-QTL-2 region from CDC Limerick, and PR-28-33, which introgressed the PR-QTL-3 region from CDC Limerick, were 5.4% and 3.5% higher in protein concentration than CDC Amarillo, respectively. These results are in alignment with what was expected, given the positive additive effect of PC-QTL-1 and negative additive effect of PC-QTL-2 and PC-QTL-3.

PR-25 consists of 108 progeny lines, which is a reasonably sized mapping population to identify the QTLs of complex traits, but a larger population size is ideal for fine mapping of the traits, based on a greater number of recombination events. Several previous pea studies used a similar or a smaller number of progeny lines without substantial compromise on QTL resolution [34,35,36]. Ideally, having a larger population would increase the power of QTL identification, and could narrow the QTLs in PR-25. It should be noted that this study is based on the measurement of a large number of recombinations within the population given that the genotyping is based on a 90K SNP array.

There are rising concerns about the impact of climate change on crop production, as it may affect food security. It is a challenge to maintain the stability of crop production under erratic weather conditions. In this study, PR-25 was grown in field trials in three consecutive years and the varied weather shifted the yield and protein production yearly. In 2019 and 2020, the temperature and precipitation were moderate during the flowering and seed development stages, which resulted in a high yield but moderate protein concentration. In 2021, plants faced substantial heat and drought stress, particularly in June and July. This resulted in a relatively low seed yield in 2021, but with a greater mean protein concentration compared to 2019 and 2020. The contradictory effects of some genetic loci on the yield and on the protein content of pea were known previously [17]. The mean yield*protein from the 2021 field trials was significantly lower compared to 2019 and 2020, indicating that the hot/dry summer, especially during the flowering and seed development stages, caused substantial decline in the total protein production, despite the increased protein concentration in the seeds.

## 5. Conclusions

Three protein-related QTLs were identified from the PR-25 population. They were found on chromosomes 2, 3 and 5 and they explained 22%, 11% and 17% of the phenotypic variation for protein concentration, respectively. SNP markers within these QTL peaks could be used for marker development to assist selection in pea-breeding.

## Figures and Tables

**Figure 1 genes-13-01531-f001:**
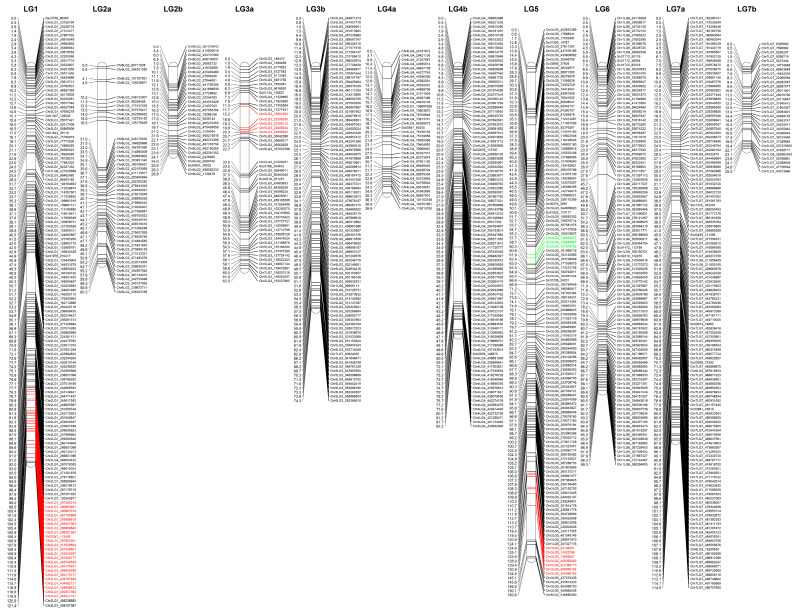
Linkage map of PR-25 population, including 11 linkage groups. A total of 139 unique loci were mapped on chromosome 2 linkage group 1 (Chr2LG1), 48 on Chr6LG2a, 31 on Chr6LG2b, 51 on Chr5LG3a, 91 on Chr5LG3b, 38 on Chr4LG4a, 97 on Chr4LG4b, 134 on Chr3LG5, 106 on Chr1LG6, 135 on Chr7LG7a, and 31 on Chr7LG7b. The total number of unique loci was 901 and the total distance of this linkage map was 855.35 cM. Three QTLs were found related to protein concentration and were located on chromosomes 2, 3 and 5 and one QTL was identified for yield which was located on chromosome 5. These protein-related QTLs (PC-QTL-1, PC-QTL-2 and PC-QTL-3) accounted for 22%, 11% and 17% of the phenotypic variation for protein concentration in PR-25, respectively. The identified QTL for yield, indicated in green, explained 11% of the phenotypic variation for yield.

**Figure 2 genes-13-01531-f002:**
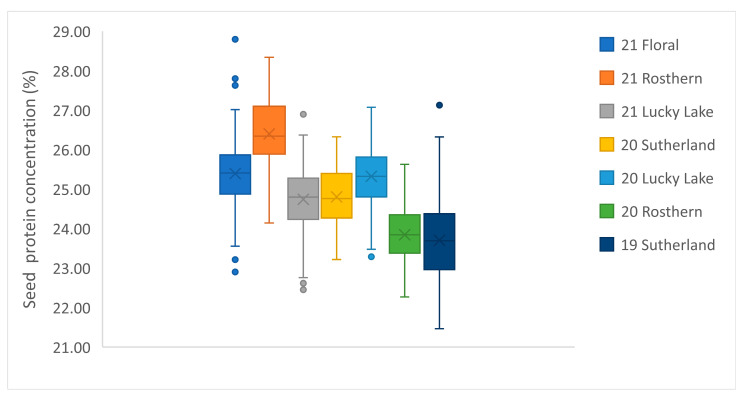
Boxplot of mean seed protein concentration of the PR-25 population (108 RILs) in 7 station-years from 2019 to 2021.

**Figure 3 genes-13-01531-f003:**
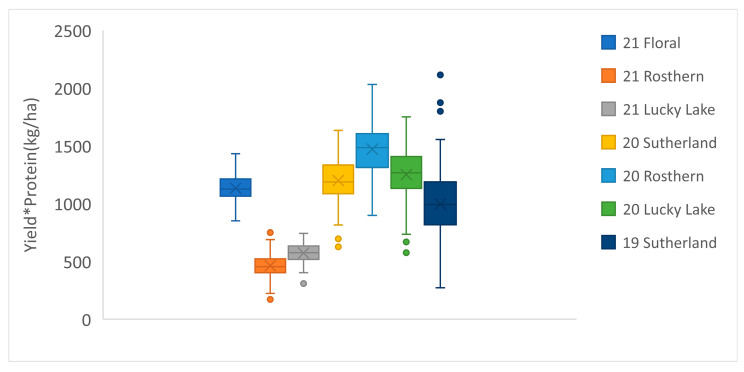
Boxplot of yield*protein of the PR-25 population (108 RILs) in 7 station-years from 2019 to 2021.

**Figure 4 genes-13-01531-f004:**
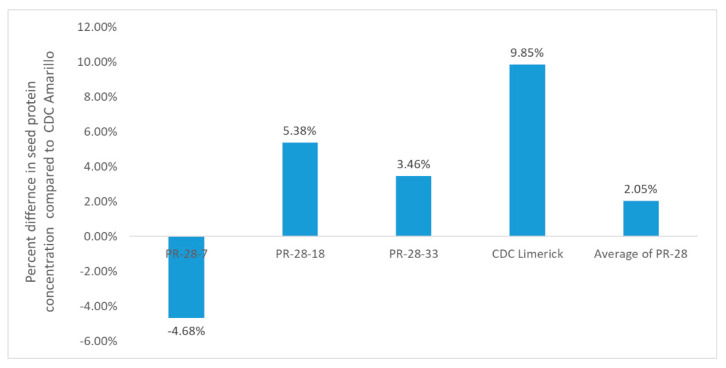
Percent difference in seed protein concentration of CDC Limerick introgression lines containing identified QTLs associated with protein concentration, and the average of all PR-28 introgression lines, compared to CDC Amarillo.

**Figure 5 genes-13-01531-f005:**
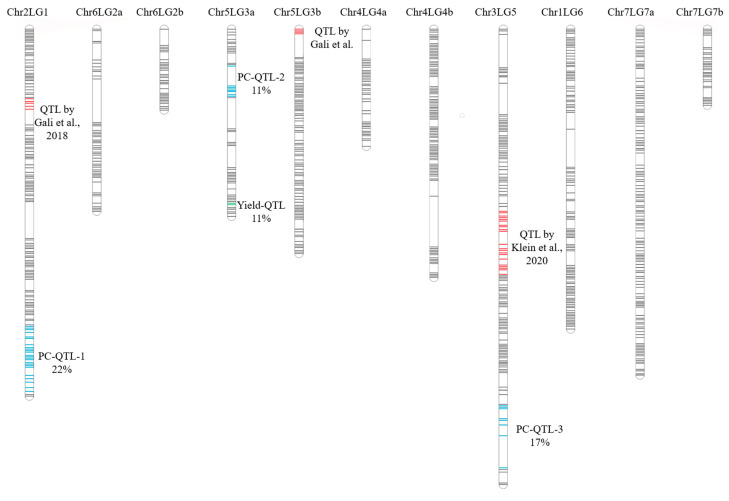
Comparison between protein concentration associated QTLs identified in this study (PC-QTL-1, PC-QTL-2, PC-QTL-3) and QTLs identified by Gali et al. [18] and Klein et al. [19].

**Table 1 genes-13-01531-t001:** Seeding date, harvest date, monthly mean temperature, and accumulated precipitation at each station-year.

Year	Station	Seeding/Harvest Date	Mean Temperature (°C)/Accumulated Precipitation (mm)				
			May	June	July	August	September
2019	Sutherland	2 May/9 September	12/20	17/182	19/136	18/26	14/74
2020	Sutherland	19 May/26 August	13/79	16/136	20/85	20/24	14/46
	Rosthern	25 May/31 August	13/79	16/136	20/85	20/24	14/46
	Lucky Lake	22 May/1 September	13/47	17/113	20/60	22/12	15/8
2021	Floral	5 May/30 August	12/61	20/64	24/38	20/66	16/4
	Rosthern	11 May/20 August	12/61	20/64	24/38	20/66	16/4
	Lucky Lake	10 May/12 August	11/63	20/76	25/45	20/61	17/2

**Table 2 genes-13-01531-t002:** Analysis of variance for protein concentration of PR-25 from 7 station-years (2019 Sutherland, 2020 Sutherland, 2020 Rosthern, 2020 Lucky Lake, 2021 Floral, 2021 Rosthern, 2021 Lucky Lake).

Source of Variation	SS	df	MS	F	*p*-Value
Block	21.65	13	1.67	2.10	0.12 ns
Genotype	792	110	7.20	9.10	<0.01 ***
Station-year	1585	6	264	333	<0.01 ***
Genotype × Environment _Interaction	774	660	1.17	1.48	<0.01 ***
Error	1130	1428	0.79		
Total	4304	2217			

Significance levels are denoted by the symbol ***, for *p* < 0.001, or not significant (ns), respectively.

**Table 3 genes-13-01531-t003:** Quantitative trait loci (QTL) for protein concentration (PC) detected using composite interval mapping (CIM) in pea recombinant inbred line population PR-25 evaluated over 7 station-years in Saskatchewan, Canada (2019–2021).

Name of QTLs	Chromosome/Linkage Group	cM Position/Peak (cM)	Flanking Markers	LOD Score	R^2^ (%)	Additive Effect
PC-QTL-1	Chr2/LG1	98.8–119.7/108.9	Chr2LG1_291265214/Chr2LG1_454521757	8.50	22	0.28
PC-QTL-2	Chr5/LG3a	13.1–22.2/20.2	Chr5LG3_15801800/Chr5LG3_23895520	4.83	11	−0.20
PC-QTL-3	Chr3/LG5	124.6–144.7/139.7	Chr3LG5_24108451/Chr3LG5_437233435	5.71	17	−0.25

Additive effects were calculated as the average performance of lines carrying A allele from CDC Amarillo minus the average performance of lines carrying B allele from CDC Limerick.

**Table 4 genes-13-01531-t004:** Pearson correlation analysis among selected agronomic traits, including seed yield, protein concentration (PC), starch, neutral detergent fiber (NDF), acid detergent fiber (ADF), Mycosphaerella disease score (Myco), days to mature (DTM), plant stand density (PSD), days to flower (DTF), height, lodging and in vitro protein digestibility (IVPD). Seven station-years’ data of PR-25, including 2019–2020 Sutherland, 2021 Floral, and 2020–2021 Rosthern and Lucky Lake, were used. All replicate data were analyzed individually.

	Yield	PC	Starch	NDF	ADF	Myco	DTM	PSD	DTF	Height
PC	−0.26 ***									
Starch	0.17 ns	−0.46 ***								
NDF	−0.19 ns	0.03 ns	−0.38 ***							
ADF	−0.29 **	0.02 ns	−0.20 *	0.51 ***						
Myco	0.02 ns	−0.09 ns	−0.11 ns	0.09 ns	−0.16 ns					
DTM	0.03 ns	0.15 ns	−0.15 ns	−0.02 ns	0.24 *	−0.49 ***				
PSD	0.32 ***	−0.09 ns	0.10 ns	−0.01 ns	−0.23 *	0.22 *	−0.14 ns			
DTF	0.11 ns	−0.03 ns	0.00 ns	−0.11 ns	−0.07 ns	−0.20 *	0.37 ***	−0.01 ns		
Height	0.52 ***	0.05 ns	0.02 ns	−0.21 *	−0.17 ns	−0.27 **	0.32 ***	0.25 **	0.44 ***	
Lodging	−0.25 **	−0.13 ns	0.06 ns	0.21 *	−0.01 ns	0.24 *	−0.23 *	0.18 ns	−0.26 **	−0.35 ***

Significance levels for the correlation coefficient (r) is denoted by the symbols *, **, ***, for *p* < 0.05, *p* < 0.01, *p* < 0.001 or not significant (ns), respectively.

**Table 5 genes-13-01531-t005:** Summary of selected CDC Limerick introgression lines that contains identified QTLs associated with protein concentration.

Information of Identified QTLs			Parents/Introgression Lines				
Name	Position	R^2^ Value	CDC Amarillo	CDC Limerick	PR-28-7	PR-28-18	PR-28-33
PC-QTL-1	Chr2	22%	+	−	−	+	+
PC-QTL-2	Chr5	11%	−	+	−	+	−
PC-QTL-3	Chr3	17%	−	+	−	−	+

Presence of corresponding QTL based on identified additive effect was denoted as “+” while absence of QTL was denoted as “−”.

## Data Availability

Data available on request from corresponding author.

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
