# Peer review of "Identification of Quantitative Trait Loci Associated with Seed Protein Concentration in a Pea Recombinant Inbred Line Population"

_genes, 2022, doi:10.3390/genes13091531_

Round 1

Reviewer 1 Report

This manuscript identified a few QTLs related to protein content in pea using a RIL population and verified them in introgression lines. Generally, this work is well performed and the results are clearly presented. I only have three minor comments list as below:

  1. Would the authors describe more clearly how they identify the SNP markers? Do you perform any high-throughput sequencing experiments, e.g. Illumina sequencing?

  2. Line 174-175, “Higher summer temperature and lower precipitation were recorded in 2021 than 2020, which resulted in lower seed yield, but greater protein concentration”. This is maybe true. But, I doubt that this statement may only derive from the author’s intuition. Maybe any supportive references added here could help.

  3. Line 226-230, Do you think only the trait performance of two introgression lines is sufficient to verify the relationship between the identified OTLs and protein content? The evidence seems weak.

Author Response

Reviewer 1 recommendations:

This manuscript identified a few QTLs related to protein content in pea using a RIL population and verified them in introgression lines. Generally, this work is well performed and the results are clearly presented. I only have three minor comments list as below:

1. Would the authors describe more clearly how they identify the SNP markers? Do you perform any high-throughput sequencing experiments, e.g. Illumina sequencing?

RESPONSE: Development of the 90K SNP array, including identification of the SNP markers used in this array, was not part of the current study. The SNP identification was based on the sequencing of a diverse panel of pea accessions and was conducted by our colleagues at INRA, France.  We obtained the SNP chip through Thermofisher for the purpose of the current research, as cited in the manuscript.  Any potential users of this SNP chip can contact Thermofisher for further information.

2. Line 174-175, “Higher summer temperature and lower precipitation were recorded in 2021 than 2020, which resulted in lower seed yield, but greater protein concentration”. This is maybe true. But, I doubt that this statement may only derive from the author’s intuition. Maybe any supportive references added here could help.

RESPONSE: In the last paragraph of the Discussion where we elaborated on this statement, we added a new sentence (line 320): “The contradictory effects of some genetic loci on yield and protein content of pea were known earlier [17]”.

3. Line 226-230, Do you think only the trait performance of two introgression lines is sufficient to verify the relationship between the identified OTLs and protein content? The evidence seems weak.

RESPONSE:  The introgression lines were chosen from a ¼-chromosome segment substitution library (CSSL) and because of this we couldn’t have had more than one line to represent the chromosome segments of each QTL region identified.  In our other ongoing projects, we are addressing this question using new mapping populations based on CDC Limerick for further validation of these QTLs.

Reviewer 2 Report

This study focused on the identification of quantitative trait loci (QTL) linked with seed protein concentration in a pea recombinant inbred line population. Three QTL have been identified and could be used in future breeding operations. The study seems to me rigorously conducted, the manuscript is very well written, and I only have few suggestions.

Line 104: ‘Trial’ should be defined.

Lines 116-117: ‘improved annually’. Improvements should be explained, along with the new correlation equation used for calculating the predicted crude protein concentration.

Figure 1: As the writing is not legible, the resolution of the figure should be increased. Moreover, the meaning of the blue and green colours should be explained in the figure legend. PC-QTL-1, PC-QTL-2 and PC-QTL-3 could be highlighted.

Figures 2 and 3: Y-axes should be labeled and units should be indicated.

Table 2: ‘GE_interactions’. The abbreviation GE should be explained.

Line 260: ‘The number of loci mapped on PR-25 linkage map 259 were higher many of the recent published pea linkage maps’. Is a word missing?

Pea: The Latin name of the species should be indicated.

Author Response

Reviewer 2 recommendations:

This study focused on the identification of quantitative trait loci (QTL) linked with seed protein concentration in a pea recombinant inbred line population. Three QTL have been identified and could be used in future breeding operations. The study seems to me rigorously conducted, the manuscript is very well written, and I only have few suggestions.

Line 104: ‘Trial’ should be defined.

RESPONSE: We replaced ‘trial’ with ‘experimental plot’

Lines 116-117: ‘improved annually’. Improvements should be explained, along with the new correlation equation used for calculating the predicted crude protein concentration.

RESPONSE: We deleted the phrase “and improved annually” and the original reference still stands accurately.   

Figure 1: As the writing is not legible, the resolution of the figure should be increased. Moreover, the meaning of the blue and green colours should be explained in the figure legend. PC-QTL-1, PC-QTL-2 and PC-QTL-3 could be highlighted.

RESPONSE: We provided a high resolution picture to add.  A separate bitmap file is submitted.  We revised the title to indicate that protein-associated QTLs are marked in red, and yield-associated QTL is marked in green.

Figures 2 and 3: Y-axes should be labeled and units should be indicated.

RESPONSE: Y-axis labels and units were added to Figures 2 and 3.

Table 2: ‘GE_interactions’. The abbreviation GE should be explained.

RESPONSE: The abbreviation ‘GE” is expanded to ‘Genotype x Environment’

Line 260: ‘The number of loci mapped on PR-25 linkage map were higher many of the recent published pea linkage maps’. Is a word missing?

RESPONSE: The sentence is modified as “The number of unique loci mapped on the PR-25 linkage map was greater than that of many of the recently published pea linkage maps”.

Pea: The Latin name of the species should be indicated.

RESPONSE: We added ‘Pisum sativum’ to the list of keywords.